# Parametric Information Bottleneck to Optimize Stochastic Neural Networks

**Thanh T. Nguyen & Jaesik Choi**
Department of Computer Engineering
Ulsan National Institute of Science and Technology
50 UNIST, Ulsan 44919, Republic of Korea
{thanhnt,jaesik}@unist.ac.kr

## Abstract

In this paper, we present a layer-wise learning of stochastic neural networks (SNNs) in an information-theoretic perspective. In each layer of an SNN, the compression and the relevance are defined to quantify the amount of information that the layer contains about the input space and the target space, respectively. We jointly optimize the compression and the relevance of all parameters in an SNN to better exploit the neural network's representation. Previously, the Information Bottleneck (IB) framework (Tishby et al. (1999)) extracts relevant information for a target variable. Here, we propose Parametric Information Bottleneck (PIB) for a neural network by utilizing (only) its model parameters explicitly to approximate the compression and the relevance. We show that, the PIB framework can be considered as an extension of the maximum likelihood estimate (MLE) principle to every layer level. We also show that, as compared to the MLE principle, PIB : (i) improves the generalization of neural networks in classification tasks, (ii) is more efficient to exploit a neural network's representation by pushing it closer to the optimal information-theoretical representation in a faster manner.

## 1 Introduction

Deep neural networks (DNNs) have demonstrated competitive performance in several learning tasks including image recognition (e.g., Krizhevsky et al. (2012), Szegedy et al. (2015)), natural language translation (e.g., Cho et al. (2014), Bahdanau et al. (2014)) and game playing (e.g., Silver et al. (2016)). Specifically in supervised learning contexts, a common practice to achieve good performance is to train DNNs with the maximum likelihood estimate (MLE) principle along with various techniques such as data-specific design of network architecture (e.g., convolutional neural network architecture), regularizations (e.g., early stopping, weight decay, dropout (Srivastava et al. (2014)), and batch normalization (Ioffe & Szegedy (2015))), and optimizations (e.g., Kingma & Ba (2014)). The learning principle in DNNs has therefore attributed to the MLE principle as a standard one for guiding the learning toward a beneficial direction. However, the MLE principle is very generic that is not specially tailored for neural networks. Thus, a reasonable question is does the MLE principle effectively and sufficiently exploit a neural network's representative power and is there any better alternative? As an attempt to address this important question, this work investigates the learning of DNNs from the information-theoretic perspective.

An alternative principle is the Information Bottleneck (IB) framework (Tishby et al. (1999)) which extracts relevant information in an input variable $X$ about a target variable $Y$. More specifically, the IB framework constructs a *bottleneck* variable $Z = Z(X)$ that is compressed version of $X$ but preserves as much relevant information in $X$ about $Y$ as possible. In this information-theoretic perspective, $I(Z, X)$ [1], the mutual information of $Z$ and $X$, captures the compression of $Z$ about $X$ and $I(Z, Y)$ represents the relevance of $Z$ to $Y$. The optimal representation $Z$ is determined via the minimization of the following Lagrangian:

$$\mathcal{L}_{IB}[p(z|x)] = I(Z, X) - \beta I(Z, Y) \tag{1}$$

---

[1] In this work, we use comma rather than semicolon to separate variables inside the mutual information operator $I(.,.)$.

where $\beta$ is the positive Lagrangian multiplier that controls the trade-off between the complexity of the representation, $I(Z, X)$, and the amount of relevant information in $Z$, $I(Z, Y)$. The exact solution to the minimization problem above is found (Tishby et al. (1999)) with the implicit self-consistent equations:

$$\begin{cases} p(z|x) & = \dfrac{p(z)}{Z(x; \beta)} \exp(-\beta D_{KL}\left[p(y|x)\|p(y|z)\right]) \\ p(z) & = \int p(z|x)p(x)dx \\ p(y|z) & = \int p(y|x)p(x|z)dx \end{cases} \tag{2}$$

where $Z(x; \beta)$ is the normalization function, and $D_{KL}[.\|.]$ is the Kullback - Leibler (KL) divergence (Kullback & Leibler (1951)). Unfortunately, the self-consistent equations are highly non-linear and still non-analytic for most practical cases of interest. Furthermore, the general IB framework assumes that the joint distribution $p(X, Y)$ is known and does not specify concrete models.

On the other hand, the goal of the MLE principle is to match the model distribution $p_{model}$ as close to the empirical data distribution $\hat{p}_D$ as possible (e.g., see Appendix I.B). The MLE principle treats the neural network model $p(\boldsymbol{x}; \boldsymbol{\theta})$ as a whole without explicitly considering the contribution of its internal structures (e.g., hidden layers and hidden neurons). As a result, a neural network with redundant information in hidden layers may have a good distribution match in a training set but show a poor generalization in test sets. In the MLE principle, we only need empirical samples of the joint distribution to maximize the likelihood function of the model given the data. The MLE principle is proved to be mathematically equivalent to the IB principle for the multinomial mixture model for clustering problem when the input distribution $X$ is uniform or has a large sample size (Slonim & Weiss (2002)). However in general the two principles are not obviously related.

In this work, we leverage neural networks and the IB principle by viewing neural networks as a set of encoders that sequentially modify the original data space. We then propose a new generalized IB-based objective that takes into account the compression and relevance of all layers in the network as an explicit goal for guiding the encodings in a beneficial manner. Since the objective is designed to optimize all parameters of neural networks and is mainly motivated by the IB principle for deep learning (Tishby & Zaslavsky (2015)), we name this method the Parametric Information Bottleneck (PIB). Because the generalized IB objective in PIB is intractable, we approximate it using variational methods and Monte Carlo estimation. We propose re-using the existing neural network architecture as variational decoders for each hidden layers. The approximate generalized IB objective in turn presents interesting connections with the MLE principle. We show that our PIBs have a better generalization and better exploit the neural network's representation by pushing it closer to the information-theoretical optimal representation as compared to the MLE principle.

## 2 RELATED WORK

Originally, the general IB framework is proposed in Tishby et al. (1999). The framework provides a principled way of extracting the relevant information in one variable $X$ about another variable $Y$. The authors represent the exact solution to the IB problem in highly-nonlinear self-consistent equations and propose the iterative Blahut Arimoto algorithm to optimize the objective. However, the algorithm is not applicable to neural networks. In practice, the IB problem can be solved efficiently in the following two cases only: (1) $X, Y$ and $Z$ are all discrete (Tishby et al. (1999)); or (2) $X, Y$ and $Z$ are mutually joint Gaussian (Chechik et al. (2005)) where $Z$ is a bottleneck variable.

Recently, the IB principle has been applied to DNNs (Tishby & Zaslavsky (2015)). This work proposes using mutual information of a hidden layer with the input layer and the output layer to quantify the performance of DNNs. By analyzing these measures with the IB principle, the authors establish an information-theoretic learning principle for DNNs. In theory, one can optimize the neural network by pushing up the network and all its hidden layers to the IB optimal limit in a layer-wise manner. Although the analysis offers a new perspective about optimality in neural networks, it proposes general analysis of optimality rather than a practical optimization criteria. Furthermore, estimating mutual information between the variables transformed by network layers and the data variables poses several computational challenges in practice that the authors did not address in the work. A small change in a multi-layered neural network could greatly modify the entropy of the input variables. Thus, it is hard to analytically capture such modifications.

The recent work Alemi et al. (2016) also uses variational methods to approximate the mutual information as an attempt to apply the IB principle to neural networks. Their approach however considers one single bottleneck and parameterizes the encoder $p(\boldsymbol{z}|\boldsymbol{x}; \boldsymbol{\theta})$ by an entire neural network. The encoder maps the input variable $\boldsymbol{x}$ to a single bottleneck variable $\boldsymbol{z}$ that is not a part of the considered neural network architecture. Therefore, their approach still treats a neural network as a whole rather than optimizing it layer-wise. Furthermore, the work imposes a variational prior distribution in the code space to approximate its actual marginal distribution. However, the variational approximate distribution for the code space may be too loose while the actual marginal distribution can be sampled easily.

Our work, on the other hand, focuses on better exploiting intermediate representations of a neural network architecture using the IB principle. More specifically, our work proposes an optimization IB criteria for an existing neural network architecture in an effort to better learn the layers' representation to their IB optimality. In estimating mutual information, we adopted the variational method as in Alemi et al. (2016) for $I(Z, Y)$ but use empirical estimation for $I(Z, X)$. Furthermore, we exploit the existing network architecture as variational decoders rather than resort to variational decoders that are not part of the neural network architecture.

## 3 PARAMETRIC INFORMATION BOTTLENECK

This section presents an information-theoretic perspective of neural networks and then defines our PIB framework. This perspective paves a way for the soundness of constraining the compression-relevance trade-off into a neural network.

We denote $X, Y$ as the input and the target (label) variables of the data, respectively; $Z_l$ as a stochastic variable represented by the $l^{th}$ hidden layer of a neural network where $1 \leq l \leq L$, $L$ is the number of hidden layers. We extend the notations of $Z_l$ by using the convention $Z_0 := X$ and $Z_{-1} := \emptyset$. The space of $X, Y$ and $Z_l$ are denoted as $\mathcal{X}, \mathcal{Y}$ and $\mathcal{Z}_l$, respectively. Each respective space is associated with the corresponding probability measures $p_D(\boldsymbol{x}), p_D(\boldsymbol{y})$ and $p(\boldsymbol{z}_l)$ where $p_D(.)$ indicates the underlying probability distribution of the data and $p(.)$ denotes model distributions. Each $Z_l$ is stochastically mapped from the previous stochastic variable $Z_{l-1}$ via an encoder $p(\boldsymbol{z}_l|\boldsymbol{z}_{l-1})$. We name $Z_l, 1 \leq l \leq L$ as a (information) bottleneck or code variable of the network. In this work, we focus on binary bottlenecks where $Z_l \in \{0, 1\}^{n_l}$ and $n_i$ is the dimensionality of the bottleneck space.

### 3.1 NEURAL NETWORKS AS SEQUENTIAL QUANTIZATION

An encoder $p(\boldsymbol{z}|\boldsymbol{x})$ introduces a soft partitioning of the space $\mathcal{X}$ into a new space $\mathcal{Z}$ whose probability measure is determined as $p(\boldsymbol{z}) = \int p(\boldsymbol{z}|\boldsymbol{x})p_D(\boldsymbol{x})d\boldsymbol{x}$. The encoding can modify the information content of the original space possibly including its dimensionality and topological structure. On average, $2^{H(X|Z)}$ elements of $\mathcal{X}$ are mapped to the same code in $\mathcal{Z}$. Thus, the average volume of a partitioning of $\mathcal{X}$ is $2^{H(X)}/2^{H(X|Z)} = 2^{I(X,Z)}$. The mutual information $I(Z, X)$ which measures the amount of information that $Z$ contains about $X$ can therefore quantify the quality of the encoding $p(\boldsymbol{z}|\boldsymbol{x})$. A smaller mutual information $I(Z, X)$ implies a more compressed representation $Z$ in terms of $X$.

Since the original data space is continuous, it requires infinite precision to represent it precisely. However, only some set of underlying explanatory factors in the the data space would be beneficial for a certain task. Therefore, lossy representation is often more helpful (and of course more efficient) than a precise representation. In this aspect, we view the hidden layers of a multi-layered neural network as a lossy representation of the data space. The neural network in this perspective consists of a series of stochastic encoders that sequentially encode the original data space $\mathcal{X}$ into the intermediate code spaces $\mathcal{Z}_l$. These code spaces are lossy representations of the data space as it follows from the data-processing inequality (DPI) (Cover & Thomas (2006)) that

$$H(X) \geq I(X, Z_l) \geq I(X, Z_{l+1}) \tag{3}$$

where we assume that $Y, X, Z_l$ and $Z_{l+1}$ form a Markov chain in that order, i.e.,

$$Y \rightarrow X \rightarrow X_l \rightarrow X_{l+1} \tag{4}$$

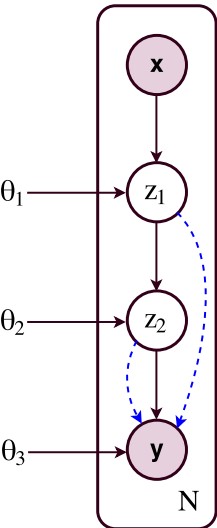

Figure 1: A directed graphical representation of a PIB of two bottlenecks. The neural network parameters $\boldsymbol{\theta} = (\boldsymbol{\theta}_1, \boldsymbol{\theta}_2, \boldsymbol{\theta}_3)$. The dashed blue arrows do not denote variable dependencies but the relevance decoders for each bottleneck. The relevance decoder $p_{true}(\boldsymbol{y}|\boldsymbol{z}_i)$, which is uniquely determined given the encoder $p_{\boldsymbol{\theta}}(\boldsymbol{z}_i|\boldsymbol{x})$ and the joint distribution $p_D(\boldsymbol{x}, \boldsymbol{y})$, is intractable. We use $p_{\boldsymbol{\theta}}(\boldsymbol{y}|\boldsymbol{z}_i)$ as a variational approximation to each intractable relevance decoder $p_{true}(\boldsymbol{y}|\boldsymbol{z}_i)$.

A learning principle should compress irrelevant information and preserve relevant information in the lossy intermediate code spaces. In the next subsection, we describe in details how a sequential series of encoders, compression and relevance are defined in a neural network.

## 3.2 PIB FRAMEWORK

Our PIB framework is an extension of the IB framework to optimize all paramters of neural networks. In neural networks, intermediate representations represent a hierarchy of information bottlenecks that sequentially extract relevant information for a target from the input data space. Existing IB framework for DNNs specifies a single bottleneck while our PIB preserves hierarchical representations which a neural network's expressiveness comes from. Our PIB also gives neural networks an information-theoretic interpretation both in network structure and model learning. In PIBs, we utilize only neural network parameters $\boldsymbol{\theta}$ for defining encoders and variational relevance decoders at every level, therefore the name *Parametric* Information Bottleneck. Our PIB is also a standard step towards better exploiting representational power of more expressive neural network models such as Convolutional Neural Networks (LeCun et al. (1998)) and ResNet (He et al. (2016)).

### 3.2.1 STOCHASTICITY

In this paper, we focus on binary bottlenecks in which the encoder $p(\boldsymbol{z}_l|\boldsymbol{z}_{l-1})$ is defined as

$$p(\boldsymbol{z}_l|\boldsymbol{z}_{l-1}) = \prod_{i=1}^{n_l} p(z_{l,i}|\boldsymbol{z}_{l-1}) \tag{5}$$

where

$$p(z_{l,i} = 1|\boldsymbol{z}_{l-1}) = \sigma(\boldsymbol{a}_i^{(l)}) = \sigma(W_{i:}^{(l)}\boldsymbol{z}_{l-1} + \boldsymbol{b}_i^{(l)}), \tag{6}$$

$\sigma(.)$ is the sigmoid function, and $W^{(l)}$ is the weights connecting the $l^{th}$ layer to the $(l+1)^{th}$ layer. Depending on the structure of the target space $\mathcal{Y}$, we can use an appropriate model for output distributions as follows: (1) For classification, we model the output distribution with softmax function, $p(Y = i|\boldsymbol{z}_L) = \mathrm{softmax}(W_{i:}^{(L+1)}\boldsymbol{z}_L + \boldsymbol{b}_i^{(L+1)})$; (2) For binary output vectors $Y$, we use a product of Bernoulli distributions, $p(\boldsymbol{y}|\boldsymbol{z}_L) = \prod_i p(\boldsymbol{y}_i|\boldsymbol{z}_L)$ where $p(Y_i = 1|\boldsymbol{z}_L) = \sigma(W_{i:}^{(L+1)}\boldsymbol{z}_L + \boldsymbol{b}_i^{(L+1)})$; (3) For real-valued output vectors $Y$, we use Gaussian distribution,

$p(Y|\boldsymbol{z}_L) = \mathcal{N}(\boldsymbol{y}; \boldsymbol{\mu} = W^{(L+1)}\boldsymbol{z}_L + \boldsymbol{b}^{(L+1)}, \boldsymbol{\sigma}^2)$. The conditional distribution $p(\boldsymbol{y}|\boldsymbol{x})$ from the model is computed using the Bayes' rule and the Markov assumption (Equation 4) in PIBs [2]:

$$p(\boldsymbol{y}|\boldsymbol{x}) = \int p(\boldsymbol{y}, \boldsymbol{z}|\boldsymbol{x})d\boldsymbol{z} = \int p(\boldsymbol{y}|\boldsymbol{z})p(\boldsymbol{z}|\boldsymbol{x})d\boldsymbol{z} = \int \prod_{l=1}^{L+1} p(\boldsymbol{z}_l|\boldsymbol{z}_{l-1})d\boldsymbol{z} \tag{7}$$

where $\boldsymbol{z} = (\boldsymbol{z}_1, \boldsymbol{z}_2, ..., \boldsymbol{z}_L)$ is the entire sequence of hidden layers in the neural network. Note that for a given joint distribution $p_D(\boldsymbol{x}, \boldsymbol{y})$, the relevance decoder $p_{true}(\boldsymbol{y}|\boldsymbol{z}_l)$ is uniquely determined if an encoding function $p(\boldsymbol{z}_l|\boldsymbol{x})$ is defined. Specifically, the relevance decoder is determined as follows:

$$p_{true}(\boldsymbol{y}|\boldsymbol{z}_l) = \int p_D(\boldsymbol{x}, \boldsymbol{y})\frac{p(\boldsymbol{z}_l|\boldsymbol{x})}{p(\boldsymbol{z}_l)}d\boldsymbol{x} \tag{8}$$

It is also important to note that many stochastic neural networks have been proposed before (e.g., Neal (1990), Neal (1992), Tang & Salakhutdinov (2013), Raiko et al. (2014), Dauphin & Grangier (2016)). However, our motivation for this stochasticity is that it enables bottleneck sampling given the data variables $(X, Y)$. The generated bottleneck samples are then used to estimate mutual information. Thus, our framework does not depend on a specific stochastic model. For deterministic neural networks, we only have one sample of hidden variables given one data point. Thus, estimating mutual information for hidden variables in this case is as hard as estimating mutual information for the data variables themselves.

### 3.2.2 LEARNING PRINCIPLE

Since the neural network is a lossy representation of the original data space, a learning principle should make this loss in a beneficial manner. Specifically in PIBs, we propose to jointly compress the network's intermediate spaces and preserve relevant information simultaneously at all layers of the network. For the $l^{th}$-level bottleneck $Z_l$, the compression is defined as the mutual information between $Z_l$ and the previous-level bottleneck $Z_{l-1}$ while the relevance is specified as its mutual information with the target variable $Y$. We explicitly define the learning objective for PIB as:

$$\mathcal{L}_{PIB}(Z) := \mathcal{L}_{PIB}(\boldsymbol{\theta}) := \sum_{l=0}^{L} \left[ \beta_l^{-1} I(Z_l, Z_{l-1}) - I(Z_l, Y) \right] \tag{9}$$

where the layer-specific Lagrangian multiplier $\beta_l^{-1}$ controls the tradeoff between relevance and compression in each bottleneck, and the concept of compression and relevance is taken to the extreme when $l = 0$ (with convention that $I(Z_0, Z_{-1}) = I(X, \emptyset) = H(X) = constant$). Here we prefer to this extreme, i.e., the $0^{th}$ level, as the *super* level. While the $l^{th}$ level for $1 \le l \le L$ indicates a specific hidden layer $l$, the super level represents the entire neural network as a whole.

The objective $\mathcal{L}_{PIB}$ can be considered as a *joint* version of the theoretical IB analysis for DNNs in Tishby & Zaslavsky (2015). However, minimizing $\mathcal{L}_{PIB}$ has an intuitive interpretation as tightening the "*information knots*" of a neural network architecture simultaneously at every layer level (including the super level). Optimizing PIBs now becomes the minimization of $\mathcal{L}_{PIB}(Z)$ which attempts to decrease $I(Z_l, Z_{l-1})$ and increase $I(Z_l, Y)$ simultaneously. The decrease of $I(Z_l, Z_{l-1})$ makes the representation at the $l^{th}$-level more compressed while the increase of $I(Z_l, Y)$ promotes the preservation of relevant information in $Z_l$ about $Y$. In optimization's aspect, the minimization of $\mathcal{L}_{PIB}$ is much harder than the minimization of $\mathcal{L}_{IB}$ since $\mathcal{L}_{PIB}$ involves inter-dependent terms that even the self-consistent equations of the IB framework are not applicable to this case. Furthermore, $\mathcal{L}_{PIB}$ is intractable since the bottleneck spaces are usually high-dimensional and the relevance encoders $p_{true}(\boldsymbol{y}|\boldsymbol{z}_l)$ (computed by Equation 8) are intractable. In the following section, we present our approximation to $\mathcal{L}_{PIB}$ which fully utilizes the existing architecture without resorting to any model that is not part of the considered neural network. The approximation then leads to effective gradient-based training of PIBs.

### 3.3 APPROXIMATE LEARNING

Here, we present our approximations to the relevance and the compression terms in the PIB objective $\mathcal{L}_{PIB}$.

---

[2]Here we use integral $\int$ even for discrete-valued variables instead of sum $\sum$ for denotation simplicity.

### 3.3.1 Approximate Relevance

Since the relevance decoder $p_{true}(\boldsymbol{y}|\boldsymbol{z}_l)$ (Equation 8) is intractable, we use a variational relevance decoder $p_v(\boldsymbol{y}|\boldsymbol{z}_l)$ to approximate it. Firstly, we decompose the mutual information into a difference of two entropies:

$$I(Z_l, Y) = H(Y) - H(Y|Z_l) \tag{10}$$

where $H(Y) = constant$ can be ignored in the minimization of $\mathcal{L}(Z)$, and

$$H(Y|Z_l) = -\int p_{true}(\boldsymbol{y}|\boldsymbol{z}_l)p(\boldsymbol{z}_l)\log p_{true}(\boldsymbol{y}|\boldsymbol{z}_l)d\boldsymbol{y}d\boldsymbol{z}_l \tag{11}$$

$$= -\int p_D(\boldsymbol{x}, \boldsymbol{y})p(\boldsymbol{z}_l|\boldsymbol{x})\log p_{true}(\boldsymbol{y}|\boldsymbol{z}_l)d\boldsymbol{z}_l d\boldsymbol{x}d\boldsymbol{y} \tag{12}$$

$$= -\int p_D(\boldsymbol{x}, \boldsymbol{y})p(\boldsymbol{z}_l|\boldsymbol{x})\log p_v(\boldsymbol{y}|\boldsymbol{z}_l)d\boldsymbol{z}_l d\boldsymbol{x}d\boldsymbol{y}$$

$$\quad -\int p(\boldsymbol{z}_l)D_{KL}[p_{true}(\boldsymbol{y}|\boldsymbol{z}_l)||p_v(\boldsymbol{y}|\boldsymbol{z}_l)]d\boldsymbol{z}_l \tag{13}$$

$$\leq -\int p_D(\boldsymbol{x}, \boldsymbol{y})p(\boldsymbol{z}_l|\boldsymbol{x})\log p_v(\boldsymbol{y}|\boldsymbol{z}_l)d\boldsymbol{z}_l d\boldsymbol{x}d\boldsymbol{y} \tag{14}$$

$$= -\mathbb{E}_{p_D(\boldsymbol{x},\boldsymbol{y})}\left[\mathbb{E}_{p(\boldsymbol{z}_l|\boldsymbol{x})}\left[\log p_v(\boldsymbol{y}|\boldsymbol{z}_l)\right]\right] =: \tilde{H}(Y|Z_l) \tag{15}$$

where the equality in Equation 12 holds due to the Markov assumption (Equation 4). In PIBs, we propose to use the higher-level part of the existing network architecture at each layer to define the variational relevance encoder for that layer, i.e., $p_v(\boldsymbol{y}|\boldsymbol{z}_l) = p(\boldsymbol{y}|\boldsymbol{z}_l)$ where $p(\boldsymbol{y}|\boldsymbol{z}_l)$ is determined by the network architecture. In this case, we have:

$$p_v(\boldsymbol{y}|\boldsymbol{z}_l) = p(\boldsymbol{y}|\boldsymbol{z}_l) = \int \prod_{i=l}^{L+1} p(\boldsymbol{z}_{i+1}|\boldsymbol{z}_i)d\boldsymbol{z}_L...d\boldsymbol{z}_{l+1} = \mathbb{E}_{p(\boldsymbol{z}_L|\boldsymbol{z}_l)}\left[p(\boldsymbol{y}|\boldsymbol{z}_L)\right] \tag{16}$$

We will refer to $\tilde{H}(Y|Z_l)$ as the *variational conditional relevance* (VCR) for the $l^{th}$-level bottleneck variable $Z_l$ for the rest of this work. In the following, we present two important results which indicate that the relevance terms in our objective is closely and mutually related to the concept of the MLE principle.

**Proposition 3.1.** *The VCR at the super level (i.e., $l = 0$) equals the negative log-likelihood (NLL) function.*

**Proposition 3.2.** *The VCR at the highest-level bottleneck variable $Z_L$ equals the VCR for the entire compositional bottleneck variable $Z = (Z_1, Z_2, ..., Z_L)$ which is an upper bound on the NLL. That is,*

$$\tilde{H}(Y|Z_L) = \tilde{H}(Y|Z) \geq -\mathbb{E}_{p_D(\boldsymbol{x},\boldsymbol{y})}\left[\log p(\boldsymbol{y}|\boldsymbol{x})\right] \tag{17}$$

While the Proposition 3.1 is a direct result of Equation 16, the Proposition 3.2 holds due to Jensen's inequality (its detail derivation in Appendix I.A).

In PIB's terms, the MLE principle can be interpreted as increasing the VCR of the network as a whole while the PIB objective takes into account the VCR at every level of the network. In turn, the VCR can also be interpreted in terms of the MLE principle as follows. It follows from Equation 15 and 16 that the VCR for layer $l$ (including $l = 0$) is the NLL function of $p(\boldsymbol{y}|\boldsymbol{z}_l)$. Therefore, increasing the Relevance parts of $J_{PIB}$ is equivalent to performing the MLE principle for every layer level instead of the only super level as in the standard MLE. Another interpretation is that our PIB framework encourages forwarding *explicit* information from all layer levels for better exploitation during learning while the MLE principle performs an *implicit* information forwarding by using only information from the super level. Finally, the VCR for a multivariate $\boldsymbol{y}$ can be decomposed into the sum of that for each component of $\boldsymbol{y}$ (see Appendix I.C).

### 3.3.2 APPROXIMATE COMPRESSION

The compression terms in $\mathcal{L}_{PIB}$ involve computing mutual information between two consecutive bottlenecks. For simplicity, we present the derivation of $I(Z_1, Z_0)$ only [3]. For the compression, we decompose the mutual information as follows:

$$I(Z_1, Z_0) = H(Z_1) - H(Z_1|Z_0) \tag{18}$$

which consists of the entropy and conditional entropy term. The conditional entropy can be further rewritten as:

$$H(Z_1|Z_0) = \int p(\boldsymbol{z}_0) H(Z_1|Z_0 = \boldsymbol{z}_0) d\boldsymbol{z}_0 = \int p(\boldsymbol{z}_0) \sum_{i=1}^{N_1} H(Z_{1,i}|Z_0 = \boldsymbol{z}_0) d\boldsymbol{z}_0$$

$$= \mathbb{E}_{p(\boldsymbol{z}_0)} \left[ \sum_{i=1}^{N_1} H(Z_{1,i}|Z_0 = \boldsymbol{z}_0) \right] \tag{19}$$

where $Z_1 = (Z_{1,i})_{i=1}^{N_1}$ and $H(Z_{1,i}|Z_0 = \boldsymbol{z}_0) = -q \log q - (1-q) \log(1-q)$ where $q = p(Z_{1,i} = 1|Z_0 = \boldsymbol{z}_0)$. For the entropy term $H(Z_1)$, we resort to empirical samples of $\boldsymbol{z}_1$ generated by Monte Carlo sampling to estimate the entropy:

$$H(Z_1) = -\mathbb{E}_{p(\boldsymbol{z}_1)}[\log p(\boldsymbol{z}_1)] \approx -\frac{1}{M} \sum_{k=1}^{M} \log p(\boldsymbol{z}_1^{(k)}) =: \hat{H}_{MLE}(Z_1) \tag{20}$$

where $\boldsymbol{z}_1^{(k)} \sim p(\boldsymbol{z}_1) = \mathbb{E}_{p(\boldsymbol{z}_0)}[p(\boldsymbol{z}_1|\boldsymbol{z}_0)]$. This estimator is also known as the maximum likelihood estimator or 'plug-in' estimator (Antos & Kontoyiannis (2001)). The larger number of samples $M$ guarantees the better plug-in entropy by the following bias bound (Paninski (2003))

$$|\mathbb{E}[\hat{H}_{MLE}(Z_1)] - H(Z_1)| \leq \log(1 + \frac{|\mathcal{Z}_1| - 1}{M}) \tag{21}$$

where $|\mathcal{Z}_1|$ denotes the cardinality of the space of variable $Z_1$. In practice, $\log p(\boldsymbol{z}_1)$ may be numerically unstable for large cardinality $|\mathcal{Z}_1|$. In the large space of $Z_1$, the probability of a single point $p(\boldsymbol{z}_1)$ may become very small that $\log p(\boldsymbol{z}_1)$ becomes numerically unstable. To overcome this problem, we propose an upper bound on the entropy using Jensen's inequality:

$$\log p(\boldsymbol{z}_1) = \log \mathbb{E}_{p(\boldsymbol{z}_0)}[p(\boldsymbol{z}_1|\boldsymbol{z}_0)] \geq \mathbb{E}_{p(\boldsymbol{z}_0)}[\log p(\boldsymbol{z}_1|\boldsymbol{z}_0)] \tag{22}$$

Thus,

$$H(Z_1) \leq -\mathbb{E}_{p(\boldsymbol{z}_1)}\left[\mathbb{E}_{p(\boldsymbol{z}_0)}[\log p(\boldsymbol{z}_1|\boldsymbol{z}_0)]\right] =: \tilde{H}(Z_1) \tag{23}$$

The upper bound $\tilde{H}(Z_1)$ is numerically stable because the conditional distribution $p(\boldsymbol{z}_1|\boldsymbol{z}_0)$ is factorized into $\prod_i p(z_{1,i}|\boldsymbol{z}_0)$, therefore, $\log p(\boldsymbol{z}_1|\boldsymbol{z}_0) = \sum_i \log p(z_{1,i}|\boldsymbol{z}_0)$ which is more stable. The upper bound $\tilde{H}(Z_1)$ can then be estimated using Monte Carlo sampling for $\boldsymbol{z}_0$ and $\boldsymbol{z}_1$.

### 3.3.3 APPROXIMATE GRADIENTS VIA BINARY BOTTLENECKS

Discrete-valued variables in PIBs make standard back-propagation not straightforward. Fortunately, one can estimate the gradient in this case. The authors in Tang & Salakhutdinov (2013) used a Generalized EM algorithm while Bengio et al. (2013) proposed to resort to reinforcement learning. However, these estimators have high variance. In this work, we use the gradient estimator inspired by Raiko et al. (2014) for binary bottlenecks because it has low variance despite of being biased. Specifically, a bottleneck $\boldsymbol{z} = (z_1, z_2, ..., z_{n_l})$ can be rewritten as being continuous by $z_i = \sigma(a_i) + \epsilon_i$ where

$$\epsilon_i = \begin{cases} 1 - \sigma(a_i) \text{ with probability } \sigma(a_i) \\ -\sigma(a_i) \text{ with probability } 1 - \sigma(a_i) \end{cases}$$

The bottleneck component $z_i$ defined as above still gets value of either $0$ or $1$ but it is decomposed into the sum of a deterministic term and a noise term. The gradient is then propagated only through the deterministic term and ignored in the noise term. A detail of gradient-based training of PIB is presented in Algorithm 1. One advantage of $GRAD\text{-}PIB$ algorithm is that it requires only a single forward pass to estimate all the information terms in $\tilde{\mathcal{L}}_{PIB}$ since the generated samples are re-used to compute the information terms at each layer level.

---

[3]The extension at the other levels is straightforward from the derivation of $I(Z_1, Z_0)$.

**Algorithm 1** Minibatch version of training PIB, we use $M = 16$ for training (and $M = 32$ for testing).

1: **procedure** GRAD-PIB
2: **Input**: Labeled training dataset $S_D$
3: $\boldsymbol{\theta} \leftarrow$ Initialize parameters
4: *repeat*:
5:     $(\boldsymbol{x}_i, \boldsymbol{y}_i)_{i=1}^N \leftarrow$ Random minibatch of $N$ samples drawn from $S_D$
6:     Generate $M$ samples of $\boldsymbol{z}_i$ per each sample of $\boldsymbol{z}_{i-1}$ for $1 \leq i \leq L$
7:     Use the generated samples above and Equations 15 and 23 to approximate $\tilde{\mathcal{L}}_{PIB}(\boldsymbol{\theta})$
8:     $\boldsymbol{g} \leftarrow \frac{\partial}{\partial \boldsymbol{\theta}} \tilde{\mathcal{L}}_{PIB}(\boldsymbol{\theta})$ using Raiko estimator
9:     $\boldsymbol{\theta} \leftarrow$ Update parameters using the approximate gradients $\boldsymbol{g}$ and SGD
10: until convergence of parameters $\boldsymbol{\theta}$
11: **Output**: $\boldsymbol{\theta}$
12: **end procedure**

## 4 EXPERIMENTS

We used the same architectures for PIBs and Stochastic Feed-forward Neural Networks (SFNNs) (e.g., Tang & Salakhutdinov (2013)) and trained them on the MNIST dataset (LeCun et al. (1998)) for image classification, odd-even decision problem and multi-modal learning. Here, a SFNN simply prefers to feed-forward neural network models following the MLE principle for learning model parameters. Each hidden layer in SFNNs is also considered as a stochastic variable. The aforementioned tasks are to evaluate PIBs, as compared to SFNNs, in terms of generalization, learning dynamics, and capability of modeling complicated output structures, respectively. All models are implemented using Theano framework (Al-Rfou et al. (2016)).

### 4.1 MNIST CLASSIFICATION

In this experiment, we compare PIBs with SFNNs and deterministic neural networks in the classification task. For comparisons, we trained PIBs and five additional models. The first model (Model A) is a deterministic neural network. In Model D, we used the weight trained in Model A to perform stochastic prediction at test time. Model E is SFNN and Model B is Model C with deterministic prediction during test phase. Model C uses the weighted trained in PIB but we report deterministic prediction instead of stochastic prediction for test performance.

| | Model | Mean (%) | Std dev. |
|---|---|---|---|
| deterministic | deterministic (A) | 1.73 | - |
| | SFNN as deterministic (B) | 1.88 | - |
| | PIB as deterministic (C) | **1.46** | - |
| stochastic | deterministic as stochastic (D) | 2.30 | 0.07 |
| | SFNN (E) | 1.94 | 0.036 |
| | PIB | **1.47** | **0.034** |

Table 1: The MNIST classification results of various models.

The MNIST dataset (LeCun (1998)) contains a standard split of 60000, and 10000 examples of handwritten digit images for training and test, respectively in which each image is grayscale of size $28 \times 28$ pixels. We used the last 10000 images of the training set as a holdout set for tuning hyperparameters. The best configuration chosen from the holdout set is used to retrain the models from scratch in the full training set. The result in the test set is then reported (for stochastic prediction, we report mean and standard deviation). We scaled the images to $[0, 1]$ and do not perform any other data augmentation. These base configurations are applied to all six models we use in this experiment.

The base architecture is a fully-connected, sigmoid activation neural network with two hidden layers and 512 units per layer. Weights are initialized using Xavier initialization (Glorot & Bengio (2010)).

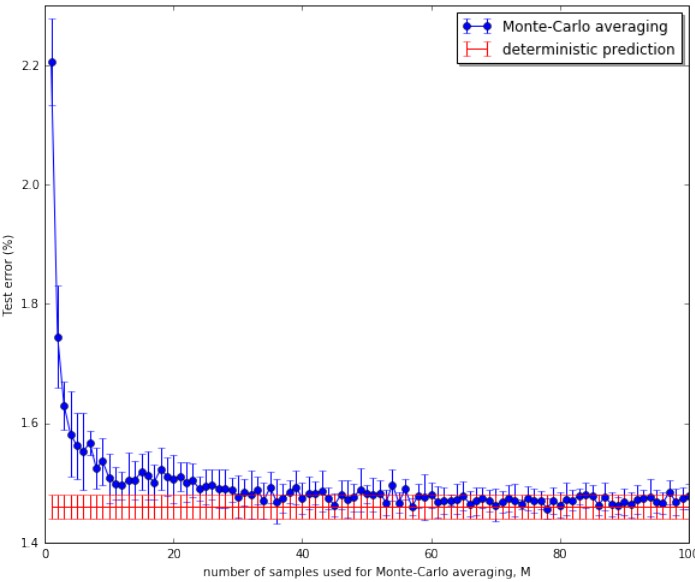

Figure 2: A comparison of Monte-Carlo averaging and deterministic prediction of PIB.

Models were optimized with stochastic gradient descent with a constant learning rate of $0.1$ and a batch size of $8$. For stochastic sampling, we generate $M = 16$ samples per point during training and $M = 32$ samples per point during testing. For stochastic prediction, we run the prediction 10 times and report its mean and deviation standard. For PIBs, we set $\beta_l = \beta, \forall 1 \leq l \leq L$. We tuned $\beta$ from $\{0\} \cup \{10^{-i} : 1 \leq i \leq 7\}$, and found $\beta^{-1} = 10^{-4}$ works best.

Table 1 provides the results in the MNIST classification error in the test set for PIB and the comparative models (A), (B), (C), (D), and (E). As can be seen from the table, PIB and Model C gives nearly the same performance which outperform deterministic neural networks and SFNNs, and their stochastic and deterministic version.

It is interesting to empirically see that the deterministic version of PIB at test time (Model C) gives a slightly better result than PIB. This also empirically holds for the case of SFNN. To investigate more in this, we compute the test error for various values of the number of samples used for Monte-Carlo averaging, $M$ (Figure 2). As we can see from the figure, the Monte-Carlo averaging of PIB obtains its good approximation around $M = 30$ and the deterministic prediction roughly places a lower bound on the Monte-Carlo averaging at test time. For visualization of learned filters of PIB, see Appendix II.A.

## 4.2 LEARNING DYNAMICS

One way to visualize the learning dynamic of each layer of a neural network is to plot the layers in the information plane (Tishby et al. (1999), Slonim (2003)). The information plane is an information-theoretic plane that characterizes any representation $Z = Z(X)$ in terms of $(I(Z, Y), I(Z, X))$ given the joint distribution $I(X, Y)$. The plane has $I(Z, X)$ and $I(Z, Y)$ as its horizontal axis and its vertical axis, respectively. In the general IB framework, each value of $\beta$ specifies a unique point of $Z$ in the information plane. As $\beta$ varies from 0 to $\infty$, $Z$ traces a concave curve, known as information curve for representation $Z$, with a slope of $\beta^{-1}$. The information-theoretic goal of learning a representation $Z = Z(X)$ is therefore to push $Z$ as closer to its corresponding optimal point in the information curve as possible. For multi-layered neural networks, each hidden layer $Z_l$ is a representation that can also be quantified in the information plane.

In this experiment, we considered an odd-even decision problem in the MNIST dataset in which the task is to determine if the digit in an image is odd or even. We used the same neural network architecture of 784-10-10-10-1 for PIB and SFNN and trained them with SGD with constant learning rate of 0.01 in the first 50000 training samples. We used three different randomly initialized neural

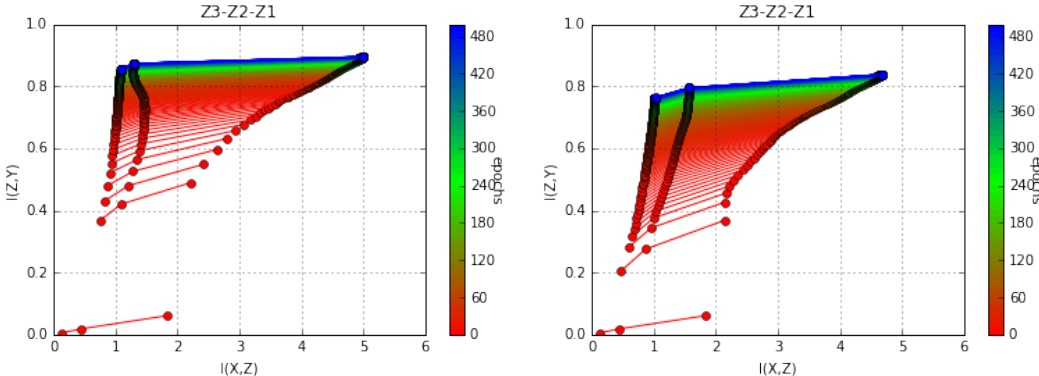

Figure 3: The learning dynamic of PIB (left) and SFNN (right) in a decision problem are presented in the information plane. Each point represents a hidden layer while the color indicate epochs. Because of the Markov assumption (Equation 4), we have $H(X) \geq I(Z_i, X) \geq I(Z_{i+1}, X)$ and $I(X, Y) \geq I(Z_l, Y) \geq I(Z_{l+1}, Y)$.

networks and averaged the mutual informations. For PIB, we used $\beta_l^{-1} = \beta^{-1} = 10^{-4}$. Since the network architecture is small, we can compute mutual information $I_x := I(Z_i, X)$ and $I_y := I(Z_i, Y)$ precisely and plot them over training epochs.

As indicated by Figure 3, both PIB and SFNN enable the network to gradually encode more information into their hidden layers at the beginning as $I(Z_i, X)$ increases. The encoded information at the beginning also contains some relevant information for the target variable as $I(Z_i, Y)$ increases as well. However, information encoding in the PIB is more selective as it quickly encodes more relevant information (it reaches higher $I(Z, Y)$ but in lesser number of epochs) while keeps the layers concise at higher epochs. The SFNN, on the other hand, encodes information in a way that matches the model distribution to the empirical data distribution. As a result, it may encode irrelevant information that hurts the generalization.

For additional visualization, an empirical architecture analysis of PIB and SFNN is presented in Appendix II.B.

## 4.3 MULTI-MODAL LEARNING

As PIB and SFNN are stochastic neural networks, they can model structured output space in which a one-to-many mapping is required. A binary stochastic variable $z_l$ of dimensionality $n_l$ can take on $2^{n_l}$ different states each of which would give a different $y$. This is the reason why the conditional distribution $p(y|x)$ in stochastic neural networks is multi-modal.

In this experiment, we followed Raiko et al. (2014) and predicted the lower half of the MNIST digits using the upper half as inputs. We used the same neural network architecture of 392-512-512-392 for PIB and SFNN and trained them with SGD with constant learning rate of $0.01$. We trained the models in the full training set of 60000 images and tested in the test set. For PIB, we also used $\beta_l^{-1} = \beta^{-1} = 10^{-4}$. The visualization in Figure 4 indicates that PIB models the structured output space better and faster (using lesser number of epochs) than SFNN. The samples generated by PIB is totally recognizable while the samples generated by SFNN shows some discontinuity (e.g., digit $2, 4, 5, 7$) and confusion (e.g., digit 3 confuses with number 8, digit 5 is unrecognizable or confuses with number 6, digit 8 and 9 are unrecognizable).

## 5 CONCLUSION

In this paper we introduced an information-theoretic learning framework to better exploit a neural network's representation. We have also proposed an approximation that fully utilizes all parameters in a neural network and does not resort to any extra models. Our learning framework offers a principled way of interpreting and learning all layers of neural networks and encourages a more

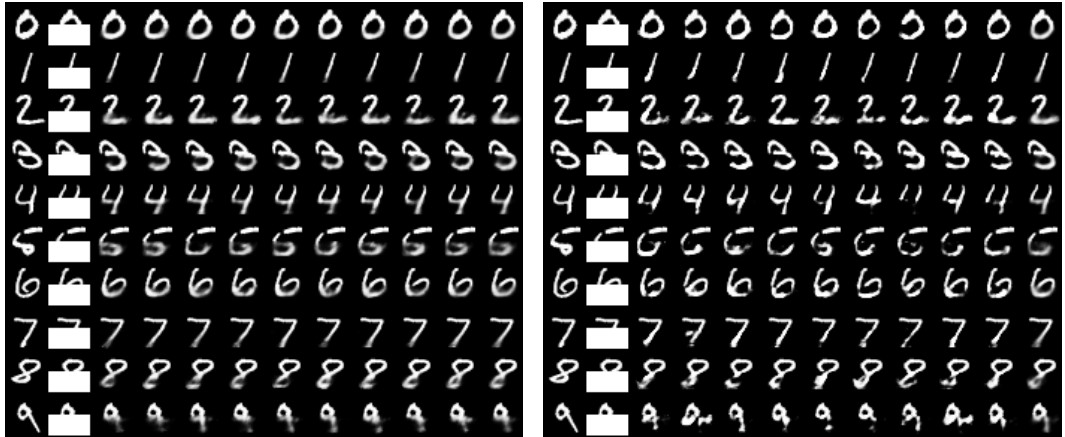

Figure 4: Samples drawn from the prediction of the lower half of the MNIST test data digits based on the upper half for PIB (left, after 60 epochs) and SFNN (right, after 200 epochs). The leftmost column is the original MNIST test digit followed by the masked out digits and nine samples. The rightmost column is obtained by averaging over all generated samples of bottlenecks drawn from the prediction. The figures illustrate the capability of modeling structured output space using PIB and SFNN.

informative yet compressed representation, which is supported by qualitative empirical results. One limitation is that we consider here fully-connected feed-forward architecture with binary hidden layers. Since we used generated samples to estimate mutual information, we can potentially extend the learning framework to larger and more complicated neural network architectures. This work is our first step toward exploiting expressive power of large neural networks using information-theoretic perspective that is not yet fully utilized.

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

## Appendix I

### A. Proof of the Prepositions

**Proof of the Preposition 3.2**: It follows from the Markov chain assumption (4) that $p(\boldsymbol{y}|\boldsymbol{z}) = p(\boldsymbol{y}|\boldsymbol{z}_L, \boldsymbol{z}_{L-1}, ..., \boldsymbol{z}_1) = p(\boldsymbol{y}|\boldsymbol{z}_L)$ and from Jensen's inequality that

$$\int p(\boldsymbol{z}|\boldsymbol{x}) \log p(\boldsymbol{y}|\boldsymbol{z}) d\boldsymbol{z} \leq \log \left( \int p(\boldsymbol{z}|\boldsymbol{x}) p(\boldsymbol{y}|\boldsymbol{z}) d\boldsymbol{z} \right) = \log p(\boldsymbol{y}|\boldsymbol{x})$$

Hence, the variational compositional relevance $\tilde{H}(Y|Z)$ equals the variational relevance for the last bottleneck and is an upper bound on the negative log-likelihood as well (Q.E.D).

### B. MLE as distribution matching

The purpose of the MLE principle can be interpreted as matching the model distribution to the empirical data distribution using the KL divergence as a measure of their discrepancy. Rigorously, given a set of samples $X = \{\boldsymbol{x}_1, \boldsymbol{x}_2, ..., \boldsymbol{x}_N\}$ i.i.d. drawn from some underlying data distribution $p_D(\boldsymbol{x})$, a parametric model $p_{model}(\boldsymbol{x}; \boldsymbol{\theta})$ attempts to map any data sample $\boldsymbol{x}$ to a real number that estimates the true probability $p_D(\boldsymbol{x})$. The MLE principle maximizes the likelihood function under the empirical data distribution. This in turn can be interpreted as matching the model distribution $p_{model}$ with the data distribution $p_D$ by minimizing their KL divergence to find the maximum likelihood (point) estimator for $\boldsymbol{\theta}$:

$$\boldsymbol{\theta}_{ML} = arg \max_{\boldsymbol{\theta}} \mathbb{E}_{\boldsymbol{x} \sim p_D(\boldsymbol{x})} \left[ \log p_{model}(\boldsymbol{x}; \boldsymbol{\theta}) \right] \tag{24}$$

$$= arg \min_{\boldsymbol{\theta}} \left[ -\mathbb{E}_{\boldsymbol{x} \sim p_D(\boldsymbol{x})} \left[ \log p_{model}(\boldsymbol{x}; \boldsymbol{\theta}) \right] + \mathbb{E}_{\boldsymbol{x} \sim p_D(\boldsymbol{x})} \left[ \log p_D(\boldsymbol{x}; \boldsymbol{\theta}) \right] \right] \tag{25}$$

$$= arg \min_{\boldsymbol{\theta}} D_{KL} \left[ p_D(\boldsymbol{x}) \| p_{model}(\boldsymbol{x}; \boldsymbol{\theta}) \right] \tag{26}$$

$$\approx arg \max_{\boldsymbol{\theta}} \sum_{i=1}^{N} \log p_{model}(\boldsymbol{x}_i; \boldsymbol{\theta}) \tag{27}$$

where expression (27) is an empirical estimation of expression (24) for N datapoints.

### C. Variational relevance for multivariate target variable

The VCR at level $l$ (defined by (15), (16)) for a multivariate variable $\boldsymbol{y}$ can be decomposed into the VCRs for each of its components. Indeed, without loss of generality, we assume bivariate target variable $\boldsymbol{y} = (y_1, y_2)$. It follows from the fact that the neurons within a layer are independent given some previous layer that we have:

$$\tilde{H}(Y|Z_l) = -\mathbb{E}_{p_D(\boldsymbol{x}, y_1, y_2)} \left[ \mathbb{E}_{p(\boldsymbol{z}_l|\boldsymbol{x})} \left[ \log p(y_1, y_2|\boldsymbol{z}_l) \right] \right] \tag{28}$$

$$= -\mathbb{E}_{p_D(\boldsymbol{x}) p_D(y_1, y_2|\boldsymbol{x})} \left[ \mathbb{E}_{p(\boldsymbol{z}_l|\boldsymbol{x})} \left[ \log p(y_1|\boldsymbol{z}_l) + \log p(y_2|\boldsymbol{z}_l) \right] \right] \tag{29}$$

$$= \sum_i -\mathbb{E}_{p_D(\boldsymbol{x}) p_D(y_i|\boldsymbol{x})} \left[ \mathbb{E}_{p(\boldsymbol{z}_l|\boldsymbol{x})} \left[ \log p(y_i|\boldsymbol{z}_l) \right] \right] \tag{30}$$

$$= \sum_i \tilde{H}(Y_i|Z_l) \tag{31}$$

