# OpenReview forum: "Parametric Information Bottleneck to Optimize Stochastic Neural Networks"
_ICLR.cc/2018/Conference — Reject_

### Official Review · AnonReviewer1 · 2017-11-27
**Exciting direction, subpar writing, insufficient experimental protocol**

**Rating:** 4
**Confidence:** 4

**Review:**

This paper proposes a learning method (PIB) based on the information bottleneck framework.
PIB pursues the very natural intuition outlined in the information bottleneck literature: hidden layers of deep nets compress the input X while maintaining sufficient information to predict the output Y.
It should be noted that the limitations of the IB for deep learning are currently under heavy discussion on OpenReview.
Optimizing the PIB objective is intractable and the authors propose an approximation that applies to binary valued stochastic networks.
They use a variational bound to deal with the relevance term, I(Z_l,Y), and  Monte Carlo sampling to deal with the layer-by-layer compression term, I(Z_l,Z_{l+1}).
They present results on MNIST aiming to demonstrate that using PIBs improves generalization and training speed.

This is a timely and interesting topic. I enjoyed learning about the authors’ proposed approach to a practical learning method based on the information bottleneck. However, the writing made it challenging and the experimental protocol raised some serious questions. In summary, I think the paper needs very careful editing for grammar and language and, more importantly, it needs solid experiments before it’s ready for publication. When that is done it would make an exciting contribution to the community. More details follow.


Comments:
1. All architectures and objectives (both classic and PIB-based) are trained using a single, fixed learning rate (LR). In my opinion, this is a red flag. The PIB objective is new and different to the other objectives. Do all objectives happen to yield their best performance under the same LR? Maybe so, but we won’t know unless the experimental protocol prescribes a sufficient range of LRs for each architecture. In light of this, the fact that SFNN is given extra epochs in Figure 4 does not mean much.
2. The batch size for MNIST classification is unusually low (8). Common batch sizes range from 64 to 1K (typically >= 128). Why did the authors make this choice? Is 8 good for architectures A through E?
3. On a related note, the authors only seem to report results from a single random seed (ie. deterministic architectures are trained exactly once). I would like to see results from a few different random seeds. As a result of comments 1,2,3, even though I do believe in the merit of the intuition pursued and the techniques proposed, I am not convinced about the main claim of the paper. In particular, the experiments are not rigorous enough to give serious evidence that PIBs improve generalization and training speed.
4. The paper needs some careful editing both for language (cf. following point) but also notation. The authors use notation p_D() in eqn (12) without defining it. My best guess is that it is the same as p_u(), the underlying data distribution, but makes parsing the paper hard. Finally there are a few steps that are not explained: for example, no justification is given for the inequality in eqn (13).
5. Language: the paper needs some careful editing to correct numerous language/grammar issues. At times it is detrimental to understanding. For example I had to read the text leading up to eqn (8) a number of times.
6. There is no discussion of computational complexity and wall-clock time comparisons. To be clear, I think that even if the proposed approach were to be slower than the state of the art it would still be very interesting. However, there should be some discussion and reporting of that aspect as well.


Minor comments and questions:
7. Mutual information is typically typeset using a semicolon instead of a comma, eg. I(X;Z).
8. Why is the mutual information in Figure 3 so low? Are you perhaps using natural logarithms to estimate and plot I(Z;Y)? If this is base-2 logarithms I would expect a value close to 1.

---

### Official Review · AnonReviewer2 · 2017-11-27
**Interesting and to my knowledge novel idea which could benefit from additional experiments.**

**Rating:** 6
**Confidence:** 4

**Review:**

This paper presents a new way of training stochastic neural network following an information relevance/compression framework similar to the Information Bottleneck. A new training objective is defined as a sum of mutual informations (MI) between the successive stochastic hidden layers plus a sum of mutual informations between each layer and the relevance variable.

The idea is interesting and to my knowledge novel. Experiments are carefully designed and presented in details, however assessing the impact of the proposed new objective is not straightforward. It would have been interesting to compare not only with SFNN but also to a model with the same architecture and same gradient estimator (Raiko et al. 2014) using maximum likelihood. This would allow to disentangle the impact of the learning mechanism from the impact of the learning objective.

Why is it important to maximise I(X_l, Y) for every layer? Does that impact the MI of the final layer and Y?

To estimate the MI between a hidden layer and the relevance variable, a multilayer generalisation of the variational bound from Alemi et al. 2016. Computation of the bound requires integration over multiple layers (equation 15). How is this achieved in practice? With high-dimensional hidden layers a Monte-Carlo estimate on the minibatch can be very noisy and the resulting estimation of MI could be poor.

Mutual information between the successive layers is decomposed as an entropy plus a conditional entropy term (eq 17). How is the conditional entropy term estimated? The entropy term is first bounded by conditioning on the previous layer and then estimated using Monte Carlo sampling with a plug-in estimator. Plug-in estimators are known to be inefficient in high dimensions even using a full dataset unless the number of samples is very large. It thus seems challenging to use mini batch MC, how does the mini batch estimation compare to an estimation using the full dataset? What is the variance of the mini batch estimate?

In the related work section, the IB problem can also be solved efficiently for meta-Gaussian distribution as explained in Rey et al. 2012 (Meta-gaussian information bottleneck).

There is a small typo in (eq 5).

---

### Official Review · AnonReviewer3 · 2017-11-27
**Interesting line of work, but the burning questions are not yet answered: would recommend for workshop publication at this stage.**

**Rating:** 4
**Confidence:** 4

**Review:**

# Paper overview:
This paper views the learning process for stochastic feedforward networks through the lens of an
iterative information bottleneck process; at each layer an attempt is made to minimise the mutual
information (MI) with the feed-in layer while maximising the MI between that layer and the presumed-endogenous variable, 'Y'.

Two propositions are made, (although I would argue that their derivations are trivially the consequence
of the model structure and inference scheme defined), and experiments are run which compare the approach to maximum likelihood estimation for 'Y' using an equivalent stochastic network architecture.

# Paper discussion:
In general I like the idea of looking further into the effect of adding network structure on the original
information bottleneck results (empirical and theoretical).  I would be interested to see if layerwise
input skip connections (i.e. between each network layer L_i and the original input variable 'X') hastened the 'compression' stage of learning e.g. (i.e. the time during which the intermediate layers minimise MI with 'X').  I'm also interested that clear examples of the information bottleneck principle in practice (e.g. CCA) are rarely mentioned.

On the other hand, I think this paper is not quite ready: it reads like work written in a hurry, and is at times hard to follow as a result.  There are several places where I think the terminology does not quite reflect what the authors perhaps hoped to express, or was otherwise slightly clumsy e.g:

* "...self-consistent equations are highly non-linear and still too abstract to be used for many...", presumably what was implied was that the original solution to the information bottleneck as expressed by Tishby et al is non-analytic for most practical cases of interest?

* "Furthermore, we exploit the existing network architecture as variational decoders rather than resort to variational decoders that are not part of the neural network architecture." -> The existing network architecture is used to provide a variational inference framework for I(Z,Y).

* "On average, 2H(X|Z) elements of X are mapped to the same code in Z." In an ideal world I would like the assumptions required for this to hold true to be a fleshed out a little here.

* "The generated bottleneck samples are then used to estimate mutual information" -> an empirical estimation of I(Z,X) would seem a very high variance estimator; the dimensionality of X is typically large in modern deep-learning problems---do you have any thoughts on how the learning process fares as this varies?  Further on you cite that L_PIB is intractable due to the high dimensionality of the bottleneck variables, I imagine that this still yields a high var MC estimator in your approximation (in practice)?  Was the performance significantly worse without the Raiko estimator?

* "In this experiment, we compare PIBs with ...." -> I find this whole section hard to read, the description of how the models relate to each other is a little difficult to follow at first sight.

* Information dynamics of learning process (Figures 3, 6, 7, 8) -> I am curious as to why you did not run the PIB for the same number of epochs as the SFNN?  I would also argue that you did not run either method as long as you should have (both approaches lack the longer term 'compression' stage whereby layers near the input reduce I(X,Z_i) as compared to their starting condition)?  This property is visible in I(Z_2,X) for PIB in Figure 3, but otherwise absent.

# Conclusion:
In conclusion, while interesting, for me the paper is not yet ready for publication.  I would recommend this work for a workshop presentation at this stage.

---

### Decision · Program_Chairs · 2018-01-29
**ICLR 2018 Conference Acceptance Decision**

**Decision:**

Reject

**Comment:**

The reviewers are in agreement that while the paper is interesting, both the clarity of presentation and experimental rigor could be improved. The committee feels this paper is not ready for publication at ICLR 2018 inits current form.